# Rapid Change in FcεRI Occupancy on Basophils After Venom Immunotherapy Induction

**DOI:** 10.3390/ijms26157511

**Published:** 2025-08-04

**Authors:** Viktoria Puxkandl, Stefan Aigner, Teresa Burner, Angelika Lackner, Sherezade Moñino-Romero, Susanne Kimeswenger, Wolfram Hoetzenecker, Sabine Altrichter

**Affiliations:** 1Department for Dermatology and Venerology, Kepler University Hospital, 4020 Linz, Austria; viktoria.puxkandl@kepleruniklinikum.at (V.P.); stefan.aigner@kepleruniklinikum.at (S.A.); teresa.burner@jku.at (T.B.); angelika.lackner_1@jku.at (A.L.); susanne.kimeswenger@jku.at (S.K.); sabine.altrichter@kepleruniklinikum.at (S.A.); 2Center for Medical Research (ZMF), Johannes Kepler University, 4020 Linz, Austria; 3Institute of Allergology, Charité-Universitätsmedizin Berlin, Corporate Member of Freie Universität Berlin, Humboldt-Universität zu Berlin, 10117 Berlin, Germany; sherezade.monino-romero@charite.de; 4Fraunhofer Institute for Translational Medicine and Pharmacology (ITMP), Immunology and Allergology (IA), 12203 Berlin, Germany; 5Clinical Research Institute for Inflammation Medicine, Medical Faculty, Johannes Kepler University, 4020 Linz, Austria

**Keywords:** venom allergy, subcutaneous injection, specific immunotherapy, hymenoptera, bee, wasp, FcεRI, IgE, basophils, basophil activation test

## Abstract

Specific venom immunotherapy (VIT) in patients with hymenoptera venom allergy (HVA) represents a well-studied approach to reduce the severity of a possible anaphylactic reaction. Currently, data on mechanisms of tolerance induction at the cellular level within the first hours of therapy are lacking. To address this, total and unoccupied high-affinity IgE receptor (FcεRI) numbers per basophil, soluble FcεRI (sFcεRI) and serum tryptase levels were measured before and after the first day of VIT induction in HVA patients. Additionally, basophil activation tests (BATs) were performed at those time points. In the early phase of VIT induction, no significant change in total FcεRI receptor density on basophils was observed, but a significant increase in unoccupied FcεRI was noticeable, predominantly in patients with high total IgE and low baseline unoccupied FcεRI density. No meaningful difference in serum tryptase levels or sFcεRI levels was observed after VIT induction. BATs showed heterogeneous results, often unchanged before and after VIT (in 47% of the cases), sometimes increased (in 40%) and only rarely decreased EC_50_ sensitivity (in 13%). Changes in the BAT EC_50_ correlated with FcεRI receptor density changes in basophils. In summary, VIT induction led to an increased ratio of unoccupied-to-total FcεRI without notable tryptase or sFcεRI serum elevation, pointing towards subthreshold cell activation with receptor internalization and recycling. However, the mostly unchanged or even increased basophil sensitivity in EC_50_ calls for further research to clarify the clinical relevance of these rapid receptor modulations.

## 1. Introduction

Specific venom immunotherapy (VIT) for patients with hymenoptera venom allergy (HVA) is a well-established approach to reduce the severity of potential anaphylactic reactions after re-exposure to the culprit insect’s venom [1,2].

In subcutaneous specific immunotherapy (SCIT), patients receive gradually increasing doses of insect venom through subcutaneous injections, starting with a very small amount. The dosage is incrementally increased until it reaches the equivalent of one to two insect stings. Depending on the patient’s medical history and the severity of the previous anaphylactic reactions, the most accelerated form of SCIT—known as the ultra-rush protocol—can be administered over just two days to quickly reach the target dose [3,4] (Materials and Methods Section 4.2). As maintenance therapy, patients receive follow-up injections every 4 to 8 weeks for at least 3 to 5 years, according to current guidelines [5].

To date, the full mechanism behind tolerance induction within the first few hours of venom immunotherapy remains unclear. Basophils and mast cells—key drivers of anaphylaxis—also likely play a central role in the early desensitization process [6,7]. Both cell types express the high-affinity IgE receptor FcεRI on their surface, making them essential components in Type-I/IgE-driven allergies [8,9]. One study has shown that within the first 6 h of the induction phase, mRNA levels of histamine receptor 2 (H2R) are upregulated in basophils. H2R-mediated ex vivo stimulation resulted in basophil activation test (BAT) inhibition [10]. However, in vivo confirmation of this pathway is still lacking. Furthermore, previous studies suggested a desensitization of basophils due to the loss of FcεRI expression in the first week(s) between VIT induction and before the first maintenance dose [11]. Plewako et al. [6] even showed a decreased number of circulating basophils during the first days of VIT. Later in the course of VIT—over the first weeks and months—antigen-specific blocking antibodies such as IgA, IgG1 and IgG4 are produced. These antibodies compete with IgE for antigen binding, thereby inhibiting IgE–antigen crosslinking [12] and contributing to long-term suppression of Type-I anaphylactic responses.

The soluble FcεRI alpha chain (sFcεRI) could also play a role in the desensitization process. This truncated molecule emerges following IgE crosslinking of the FcεRI of effector cells after antigen exposure. It is believed to be part of a negative feedback loop, acting as an endogenous inhibitor of FcεRI on basophils and mast cells, thereby preventing further IgE binding [13,14]. Patients with higher serum sFcεRI levels (>2 ng/mL) have been shown to better tolerate drug desensitization protocols [15].

The basophil activation test (BAT) is an ex vivo test that measures the degree of degranulation of basophils following allergen stimulation. It is currently considered to be the gold standard for evaluating hymenoptera venom allergy/anaphylaxis [16,17]. By assessing basophil reactivity, the BAT provides insights into a patient’s clinical sensitivity to allergen exposure and is, therefore, regarded as a useful tool for monitoring the desensitization process during venom immunotherapy [18]. However, it has been reported that after successful VIT (confirmed by sting provocation), the BAT showed inconsistent results [16]. Arzt et al. [19] reported significantly reduced BAT activity in patients who underwent vespid VIT, but not bee VIT, compared with hymenoptera-allergic patients without VIT. Additionally, there are reports that the BAT only decreased after 6 months of VIT; before that, activation seems to be comparable with before VIT [20]. Notably, data on immediate changes in BAT responses—within just a few hours following VIT induction—are lacking in the current literature.

Despite the high efficacy of VIT [1,2], a small percentage of patients do not achieve protection by this therapy. Also, a small proportion of patients encounter significant side effects, up to severe anaphylactic reactions, upon therapy initiation. Currently, we cannot reliably identify patients at risk of severe reactions or therapy failure because we do not completely understand the biological processes at the early stages of VIT.

Therefore, we aimed to investigate if the induction of hymenoptera VIT leads to alterations in the surface expression levels of FcεRI on effector cells to create an immediate change in the release of sFcεRI and changes in BAT responses in order to gain insights into the modulation of allergic effector mechanisms in the very early stages of VIT.

## 2. Results

### 2.1. VIT Induction Is Well-Tolerated

Of the 19 patients (for clinical features, see Materials and Methods Section 4.2) undergoing ultra-rush or cluster VIT, 18 tolerated the protocol well. Local reactions at the injection site were common (78%). Blood samples were collected before and immediately after the final dose on the first day of VIT induction for further analysis.

During VIT induction, one patient (undergoing wasp-venom rush VIT induction) reported a sensation of facial swelling, the feeling of a lump in the throat and chest tightness on the third day (after 10 µg wasp venom). No objective signs of oral or facial swelling were observed and vital parameters remained within normal limits. Serum tryptase (BST) did not exceed baseline levels during this episode. Consequently, up-dosing was continued the next day with the last tolerated dose (8 µg). The subsequent increasing doses were well-tolerated until reaching the maintenance dose (Appendix A). Due to elevated BST (21.1 µg/L), further diagnostic testing was performed. Hereditary alpha tryptasemia (HaT) (TPSAB1 triplication; Genotype baa:baa) was detected, but no cKIT mutation.

Overall, serum tryptase levels did not significantly change during VIT induction (Table 1).

### 2.2. VIT Induction Reduces IgE Occupancy on Basophils but Does Not Increase sFcεRI Serum Levels

The mean total FcεRI density on basophils slightly decreased by 3% following VIT although this was not statistically significant. However, the mean unoccupied FcεRI significantly increased by an average of 21% during VIT and the ratio of unoccupied/total FcεRI also increased from a median of 1.69% to 2.25% (Table 1; individual changes are depicted in Figure 1a–c).

**Table 1 ijms-26-07511-t001:** Values before and after VIT.

*n* = 19	Before VIT	After VIT Induction	*p*-Value
**Tryptase (µg/L)**	5.6 (4.5)	5.5 (4.6)	0.07
**Total FcεRI**	215,426.0 ± 124,803.3	208,180.9 ± 130,439.8	0.17
**Unoccupied FcεRI**	5360.2 ± 4588.1	6028.2 ± 4163.8	0.04 *
**Ratio of unoccupied/total FcεRI**	0.017 (0.06)	0.023 (0.06)	<0.01 **
**sFcεRI (ng/mL)**	0.58 (0.3)	0.47 (0.28)	<0.01 **

Statistical analysis was performed with a paired *t*-test or Wilcoxon test (if abnormally distributed). Normally distributed values are depicted as median ± standard deviation; abnormally distributed values are displayed as median (interquartile range). Total and unoccupied FcεRI are expressed as receptors per basophil. VIT—venom immunotherapy; sFcεRI—soluble FcεRI. Significant difference before and after VIT is indicated as * (*p*-value <0.5) or ** (*p*-value < 0.01).

**Figure 1 ijms-26-07511-f001:**
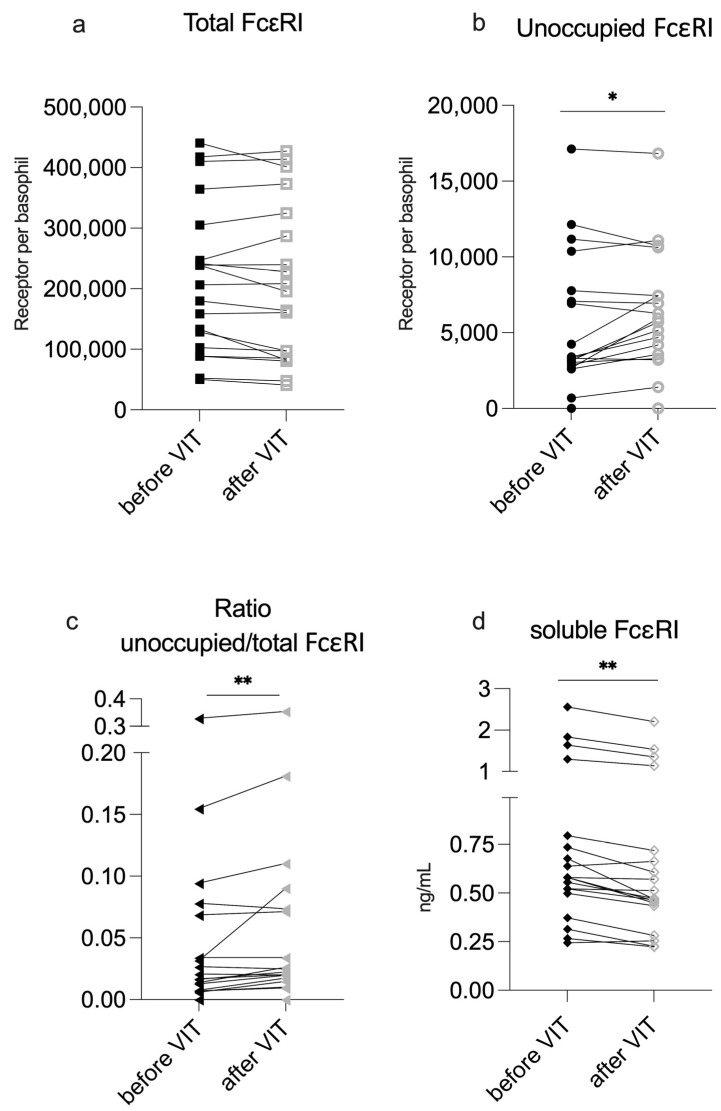
Changes after VIT Induction. Total and unoccupied FcεRI are shown as receptors per basophil. sFcεRI is shown as ng/mL. VIT—venom immunotherapy. Significant difference before and after VIT is indicated as * (*p*-value < 0.5) or ** (*p*-value < 0.01).

When grouping the patients regarding the desensitized venom, significant results comparable with the overall cohort were seen in the wasp-sensitized group, while only a trend was observed in the bee-sensitized group (Appendix A). No difference was observed regarding the change in FcεRI before and after induction when comparing the different induction protocols.

Serum sFcεRI levels were very low overall and did not correlate with total IgE levels (Appendix A). Only one of the patients presented with predefined clinically relevant levels (>2 ng/mL) [15]. Surprisingly, following VIT induction, the median sFcεRI concentration in patient sera was not elevated; instead, it was slightly, but significantly, reduced (Table 1; Figure 1d).

### 2.3. Significant Changes in Unoccupied FcεRI Density Is Primarily Seen in Patients with Low Baseline Numbers of Unoccupied FcεRI on Their Basophils

In our cohort, about half of the patients (53%) presented with a very low average number (<2%) of unoccupied FcεRI on their basophils. These patients had a significantly higher total serum IgE, but not specific IgE (Table 2). There was no difference between them in their sensitization to the culprit venom.

When dividing our cohort into “High” (over 2%) and “Low” (under 2%) unoccupied FcεRI groups, a significant increase (unoccupied FcεRI before VIT minus after VIT = mean −1222.43 receptors per basophil) in unoccupied FcεRI was observed; the ratio of unoccupied/total was only observed within the “Low” group (Table 3; Figure 2b). This indicates that more unoccupied FcεRI was present in the ‘Low’ group after VIT.

Further, this group showed a minimal, but significant, decrease in tryptase levels before and after VIT. However, a minimal decrease in soluble FcεRI was noticeable in both groups.

### 2.4. BAT Sensitivity Does Not Significantly Change During VIT Induction for Most Patients

In the BAT, wasp-sensitized individuals showed a higher mean percentage of maximum-activated basophils (mean >80% activated basophils) compared with bee-sensitized individuals (mean <60% activated basophils) (*p* = 0.07) (Appendix A).

Changes greater than 10% in the maximum-activated basophils from the BAT were considered to be potentially meaningful and were observed in 3 (15%) of the 19 patients. In two patients (one allergic to wasps, one to bees) the maximum activation increased and the maximum activation decreased only in one (wasp-sensitive) patient. The remaining patients showed only minimal changes in the number of maximally activated basophils (Appendix A).

We did not observe a significant change in overall BAT sensitivity when assessing the venom concentration at the EC_50_ level during VIT induction (Figure 3a). In 7 out of 15 patients, EC_50_ changes of more than 10% were not detectable. Only two patients (13%) had a detectable decrease in BAT EC_50_, while six patients (40%) exhibited an increase in EC_50_ (Figure 3b). An increased basophil sensitivity during wasp VIT was observed in 3 out of 4 cases. These patients initially showed a high threshold for venom concentration. Although sensitivity increased by more than 40% in some cases, these changes remained within the same order of magnitude of wasp venom dilution (Appendix A). Only one patient undergoing bee VIT demonstrated a decrease in EC_50_ of about 24%, and shifted the venom dilution category to the highest venom concentration (Appendix A).

### 2.5. Changes in BAT Sensitivity Correlate with Total FcεRI Expression on Basophils

The shift in EC_50_, predominantly towards increased basophil sensitivity, indicated by a lower venom concentration required to reach half-maximal activation, showed a highly significant correlation with the overall change in total FcεRI expression (Table 4). Despite the significance, total FcεRI changes after VIT induction in the wasp-allergic patient group were mostly subtle, except for one patient with a strong increase who also had a marked decrease in EC_50_ sensitivity (Table 4; Appendix A). A trend towards a correlation with lower total FcεRI in patients with decreased EC_50_ sensitivity was also seen in the bee-venom-allergic group (Table 4; Appendix A). When looking at the relative change in EC_50_ before and after VIT, the correlation of EC_50_ with total FcεRI was significant in the bee VIT group and trending in the wasp VIT group (Appendix A; Appendix A).

Regarding the correlation of the BAT EC_50_ with unoccupied FcεRI density, the picture was mixed. In the wasp VIT group, increased unoccupied FcεRI positively correlated with EC_50_ (Table 4; Appendix A), whereas in the bee VIT group, an increase in unoccupied FcεRI was seen in all patients, which negatively correlated with the relative EC_50_ (Appendix A; Appendix A).

## 3. Discussion

This study demonstrates that FcεRI occupancy on basophils can rapidly change within the first hours of hymenoptera VIT induction.

Individuals undergoing ultra-rush or cluster VIT induction protocols did not experience any major complications. These initiation protocols, therefore, appear to be highly reliable, as reflected in the literature [5,21]. Only one patient undergoing the ultra-rush induction protocol (with diagnosed HaT) reported subjective, but not measurable systemic symptoms during initiation; however, they were still able to complete VIT. HaT has been associated with severe anaphylactic reactions [22], although up to now, no issues have been reported for the initiation of VIT in individuals with HaT. Nevertheless, elevated BST has been reported to be a risk factor for severe reactions during VIT initiation [23,24]. In mastocytosis, which is associated with severe anaphylaxis [25,26], especially upon contact with hymenoptera venom, VIT is still recommended as the treatment of choice (in addition to an emergency kit) [5,27].

After the first day of VIT, a significant increase of over 20% in the mean unoccupied FcεRI (reduced IgE occupancy on basophils) was observed. The mean total FcεRI showed a minimal, though not significant, difference, resulting in a highly significant difference in the ratio of unoccupied-to-total FcεRI before and after VIT induction. Čelesnik et al. [11] also reported fewer FcεRI on basophils after VIT induction; however, in that study, the examination did not take place until at least five days after induction. The tryptase levels did not significantly change during this period, suggesting no significantly increase in MC degranulation.

It has been shown that sFcεRI can block the binding of IgE to free FcεRI, potentially reducing the activation of basophils and MCs, and patients with higher serum sFcεRI levels (>2 ng/mL) were less prone to reactions during desensitization [14]. Surprisingly, we did not observe an increase in sFcεRI levels, as previously described [15]. Instead, we observed a significant decrease in sFcεRI a few hours after VIT induction. However, the serum sFcεRI levels in our cohort were generally very low, similar to those of non-atopic individuals where changes are less pronounced overall. The observed decrease of 0.11 ng/mL may be biologically irrelevant, given that previous desensitization studies have reported changes of several ng/mL [28]. The reduced levels of sFcεRI and tryptase in the serum may simply be a result of the saline drip that patients receive during the process of tolerance induction for safety-measure reasons. According to the literature, a stable tryptase level is expected in patients tolerating VIT [29].

An increase in the ratio of unoccupied-to-total FcεRI without a meaningful tryptase elevation or an increase in sFcεRI may suggest a subthreshold activation of cells with IgE receptor complex internalization [30,31] and FcεRI recycling to the cell surface [32]. However, our study does not present direct experimental evidence to support this mechanism. De novo expression of FcεRI is assumed to take several hours after allergen exposure [33], so this seems an unlikely explanation for the rapid change. The suggested mechanism is supported by the results of the detailed analysis, revealing that the increase in unoccupied FcεRI was more pronounced in the “Low” (<2%) unoccupied FcεRI baseline group. This group also had higher total IgE levels overall, indicating a possible role for IgE-mediated processes. Previous studies have shown a significant correlation between unoccupied FcεRI and circulating IgE levels, supporting the notion that FcεRI expression is modulated by IgE levels [34]. Other mechanisms—such as histamine 4 receptor activation, which has been shown to regulate surface FcεRI expression in MCs—could also play a role, specifically in an autocrine feedback loop [35]. Other previously described studies focused on mechanisms such as the rapid upregulation of histamine receptor 2 on basophils within the first six hours of the VIT build-up phase. However, these studies did not examine changes in FcεRI expression on basophils in relation to suppressed FcεRI-mediated cell activation and mediator release [10]. Nevertheless, it is conceivable that unoccupied FcεRI may play a role in monitoring the efficacy of VIT.

During VIT induction, the median BAT sensitivity (max. activation and EC_50_) did not significantly change. Surprisingly, an increase in BAT sensitivity with an EC_50_ occurring at lower venom concentrations was more frequent (40%) than a decrease (13%), which would have been expected in tolerance induction. An increase in sensitivity correlated with an increase in total FcεRI expression on basophils and vice versa. However, the overall higher numbers of unoccupied FcεRI did not stringently correlate with an increase or decrease in sensitivity. In addition to IgE–FcεRI-mediated basophil activation, other mechanisms could increase the expression of surface activation markers (CD63 and CD203c) during the BAT, which could result in an earlier EC_50_ being reached at lower venom concentrations. The main question remains whether basophils play a central role in the induction of tolerance, or if MCs are the key players in both anaphylaxis and tolerance induction. Unfortunately, the tissue residency of these cells makes studies on them difficult.

A main limitation of the study is the small number of patients, and further subgrouping reduces the statistical power even more, affecting the generalizability of the results. Therefore, studies with a greater sample size are needed.

Another major limitation is the heterogeneity of the SIT protocols in this trial. Ideally, further studies should include basophil count in peripheral blood, which was not assessed in this study, as prior findings indicate a reduction in basophil numbers during the build-up phase of immunotherapy [6,10]. Furthermore, the aim should also be to assess MCs and MC-derived markers in these patients in order to determine if the observed phenomena have a further clinical impact. Understanding these mechanisms would help to identify the low percentage of patients at risk of tolerance induction failure.

## 4. Materials and Methods

### 4.1. Patients

This study analyzed patients with suspected hymenoptera venom allergy who presented at the allergy outpatient clinic of the Department of Dermatology and Venereology, Comprehensive Allergy Centre, Kepler University Hospital, between May 2022 and May 2023, and who received specific hymenoptera venom immunotherapy. In total, 19 patients agreed to participate in this study.

Ethical approval was obtained from the local ethics committee (ECS No. 1026/2022). All patient records were handled in a pseudonymized manner, following data protection and local ethics considerations. Patient data on clinical data correlation are published elsewhere.

### 4.2. Clinical and Laboratory Assessments

Patient data, including age, sex, documented allergies and concomitant diseases, were obtained from the patient charts. Hymenoptera venom allergies were graded according to Ring and Messmer’s criteria [36].

Patients were screened for mastocytosis, which was defined as prevalent if patients had a KIT-D816V mutation (and elevated tryptase >11.4 µg/L) [37]. None of the patients met the criteria for systemic mastocytosis. Additionally, each patient was screened for hereditary alpha tryptasemia (HaT) using whole blood samples at an external laboratory (MVZ Martinsried GmbH, Planegg, Germany; https://www.medicover-diagnostics.de/leistungsverzeichnis/humangenetik/hereditare-alpha-tryptasamie-hat-labor-diagnostik (accessed on 3 August 2025)) via digital droplet PCR (ddPCR) [38]. HaT was diagnosed if there was evidence of a TPSAB1 gene number variation, such as an additional copy of the gene.

All patients underwent either the ultra-rush or cluster VIT induction protocol (bee or wasp venom) (Figure 4). On the first day of the ultra-rush protocol, patients received seven subcutaneous injections every thirty minutes, starting with a concentration of 0.01 µg of the respective venom. Patients undergoing the cluster VIT protocol only received four injections on the first day of induction, starting with a venom concentration of 5 µg. The patient’s blood was first drawn immediately before the first SCIT injection (before) and after the last injection on the first day (after); this was three hours for the ultra-rush protocol and one-and-a-half hours for the cluster protocol. A cumulative dose of 151.11 µg of the respective venom was reached at that point during the ultra-rush induction, or 55 µg during the cluster induction. Only one patient underwent the rush induction protocol (Appendix A). The decision for each desensitization protocol was made according to the patient’s history and preference.

Total IgE and tryptase serum levels were assessed in the central nuclear laboratory of the Kepler University Hospital, Linz, Austria, using the ImmunoCAP System^®^ (Phadia Laboratory Systems, Thermo Fisher Scientific Inc, Uppsala, Sweden).

The characteristics of the patients are shown in Table 5.

### 4.3. Patient Grouping

Patients were grouped according to desensitized hymenoptera and protocols. The percentage of unoccupied FcεRI was calculated for each patient based on their FcεRI levels prior to VIT induction. In a further analysis, patients were grouped into unoccupied FcεRI ‘High’ and ‘Low’ groups, where the ‘High’ group was defined as an amount of unoccupied FcεRI over two percent and the ‘Low’ group had correspondingly lower values (Figure 5).

### 4.4. Basophil Activation Test (BAT) and Flow Cytometry

A BAT was performed using the allergen to which the patient was known to be sensitized, both before and after VIT induction. Double-sensitized patients received a BAT with only the allergen undergoing VIT.

In the study, a standardized BAT kit from EXIBO© (BasoFlowEx Kit Ref. ED7043) was used with bee and wasp allergens obtained from ALK (Hørsholm, Denmark) (100 µg/mL ALK wässerig SQ© 801 Bienengift and 100 µg/mL ALK wässerig SQ© 802 Wespengift). The respective venom dilutions with the ALK diluent were produced by serial dilution at the following concentrations: 1, 10^−1^, 10^−2^, 10^−3^, 10^−4^ and 10^−5^. Patients’ heparinized whole blood was stimulated with serial dilutions of the venom and processed according to the basophil test kit instructions. The probes were analyzed by flow cytometry (Beckman coulter DxFlex©, Brea, CA, USA). The gating strategy used is shown in Appendix A. A positive control used monoclonal anti-IgE from the basophil test kit.

A positive and negative control was performed for each case. As there was no cut-off given for the negative control in the manufacturer manual [39], and no generally accepted cut-offs are established for the BAT, it has been suggested that the cut-off for the negative control and its coefficient of variation should be adopted by each laboratory [40]. In the study lab, the mean ± SD of the negative control (max. activation) was 13.5% ± 8.5% and, therefore, the cut-off was set at 21%. If the negative control of the BAT reached ≥21% activation, the patient was excluded from further analyses due to a too-high preactivation.

The proposed cut-off for the positive control in the manufacturer manual [39] was given as >20%. However, as our laboratory’s set negative control was higher and the study aimed at following changes in the BAT over time (not just identifying sensitization), the cut-off was set higher, at 40%, to better monitor potentially meaningful changes. Overall, two patients were excluded due to negative control exclusions and two due to positive control exclusions.

With the created FACS data, an MS Excel calculation tool was implemented to calculate the dilution interval at which the maximal and half-maximum basophil activation (EC_50_) occurred. Furthermore, it allowed the allergen concentration at half-maximum and maximum basophil activation levels to be calculated. The highest activation level reached throughout the serial dilution was marked and showed the maximum activation level. For the MS Excel calculation tool, the activation level was plotted on the y-axis and the serial dilution on the x-axis. The different activations at each serial dilution point were connected with lines. A horizontal bar was then drawn for the half-maximum activation. The two dilution points that created a linear line, which crossed this half-maximum activation bar, were used in order to calculate the half-maximum activation via a linear function. Due to observed fluctuations at high basophil activation levels, the concentration at which maximum activation was first achieved in the BAT was determined with a 10% range tolerance. The highest activation was used as the reference point and the concentration at which >90% of the highest activation was observed was used as the calculated value for maximum BAT activation. As no meaningful differences for the BAT have been established in the literature, we considered a change of more than 10% to be potentially meaningful.

EC_50_ is given as the calculated concentration (µg/mL) at which the half-maximum activation of basophils is reached, or as a concentration step/range in which the respective calculated EC_50_ falls within [<1:10,000, 1:1000–1:10,000, 1:100–1:1000, 1:10–1:100, or 1–1:10]. A change in a shift of one concentration step (10-1) was considered to be meaningful.

### 4.5. FcεRI Expression of Basophil Profile Measurements

To evaluate FcεRI density on basophil granulocytes, patients’ heparinized whole blood was first blocked with human immunoglobulin (50g/L IG VENA (Kedrion Biopharma S.p.A., Bologna, Italy)), of which 1 µL IgG was diluted in 500 µL PBS. Anti-human CD193-APC (5E8; BD. Ref. 558208) and anti-human CD123-PE (9F5; BD. Ref. 555644) were used to identify basophils. FcεRI on basophils were stained with CRA1-BV421 (334624; BioLegend^®^, San Diego, CA, USA) for total FcεRI and CRA2-FITC (GTX00853; GeneTex©, Inc., Irvine, CA, USA GeneTex) for unoccupied FcεRI antibodies. The isotype control antibodies were IgG2b-BV421 (MPC-11; BioLegend^®^, San Diego, CA, USA. Ref. 400307) and IgG1-FITC (R&D Systems, Minneapolis, MN, USA, IC002F). The gating strategy is shown in Appendix A. MFI of CRA1 (total FcεRI) and CRA2 (unoccupied FcεRI) were assessed.

Furthermore, quantification beads (QuantumTM Simply Cellular^®^ anti-Mouse IgG; Bangs Laboratories, Inc., Fishers, IN, USA. Cat.#: 815B) were used together with the provided calculation tool (QuickCal^®^ v 2.3) to transform MFI into the “receptors per basophil granulocyte” unit. In total, seven receptor quantifications via beads and FACS were performed parallel to the acquisition of patient FACS, representing data over a timespan of about one year. The mean of these seven bead measurements was taken to transform MFI into receptors/cell. Five patients showed a slightly negative MFI in the FACS, which could not be transformed into receptors/cell with the provided calculation tool; therefore, the receptors/cell value was defined as zero for a close and standardized approximation.

### 4.6. Soluble FcεRI Measurement

After clotting and centrifugation, the patient serum was aliquoted and stored frozen at −20°C. An ELISA Kit (Invitrogen; Thermo Fisher Scientific Inc., Waltham, MA, USA. Ref.: BMS2101-2) was used for the measurement of soluble FcεRI in patient sera according to the manufacturer’s protocol. Using the provided standard, the sample duplicates were converted into concentration values (given as ng/mL).

### 4.7. Statistical Analyses

Statistical analyses were performed using ISM SPSS Statistics (Version 30.0.0.0). Normal distribution was determined by the Shapiro–Wilk test. Variables that were normally distributed were given as the median ± SD, and others as the median and interquartile range (IQR). A paired comparison was performed using a paired *t*-test (if normally distributed) or a Wilcoxon test (if abnormally distributed). A Mann–Whitney U test was used for unpaired group comparisons if they were abnormally distributed. Correlations were calculated using Spearman’s rho. A *p*-value ≤ 0.05 was considered to indicate statistical significance.

## 5. Conclusions

VIT induction was tolerated well and resulted in a measurable reduction in IgE occupancy on basophils without a corresponding increase in sFcεRI serum levels. Significant changes in the density of unoccupied FcεRI levels were primarily observed in patients with low baseline levels of unoccupied receptors, indicating a differential immunological response based on initial receptor status. Although BAT sensitivity was inconsistent during the induction phase for most patients, individual variations in BAT sensitivity correlated with changes in FcεRI surface expression.

These early immunological shifts highlight the dynamics of effector cell regulation during VIT, suggesting that FcεRI expression patterns could serve as both mechanistic markers and potential tools for monitoring treatment responses in the management of hymenoptera venom allergy.

## Figures and Tables

**Figure 2 ijms-26-07511-f002:**
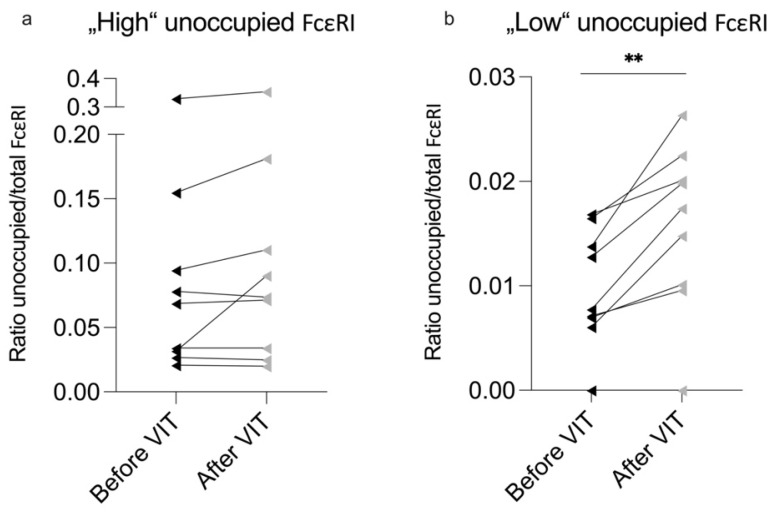
Comparison of ‘High’ and ‘Low’ unoccupied FcεRI groups. The ratio of total-to-unoccupied FcεRI in the (**a**) ‘High’ (over 2%) and (**b**) ‘Low’ (under 2%) unoccupied FcεRI groups. VIT—venom immunotherapy. Significant difference before and after VIT is indicated as ** (*p*-value < 0.01).

**Figure 3 ijms-26-07511-f003:**
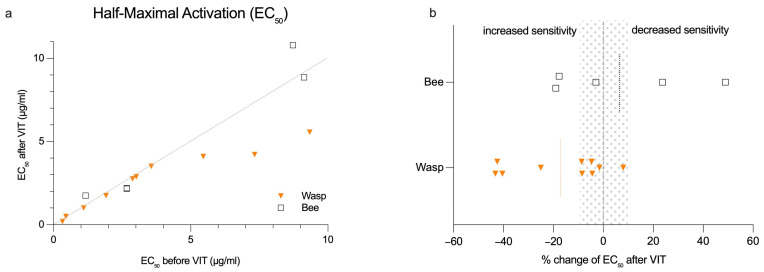
Changes in half-maximal activation (EC_50_) for BATs before/after VIT induction in bee- and wasp-allergic patients. (**a**) Absolute change in EC_50_ for BATs; (**b**) relative change in EC_50_ for BATs. Orange and black dotted line represent the mean. Gray dotted area represents ±10%.

**Figure 4 ijms-26-07511-f004:**
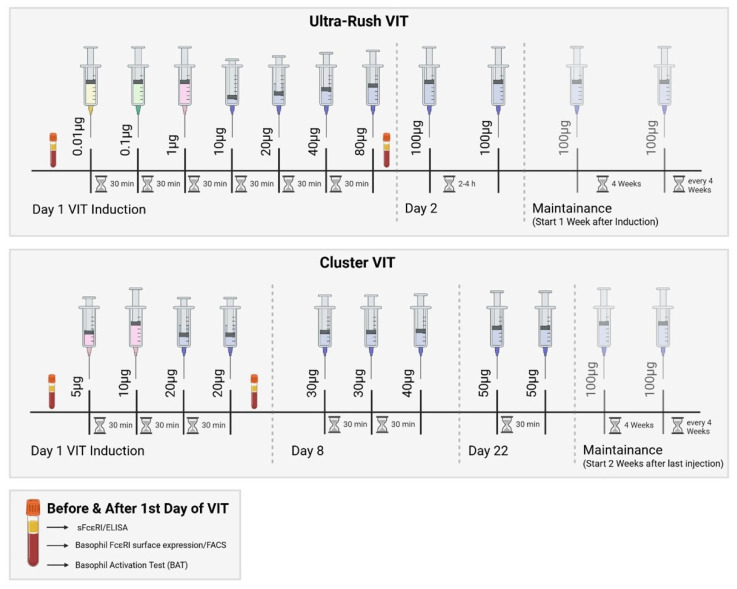
Ultra-rush and cluster VIT induction protocols with the corresponding maintenance doses. SCIT—subcutaneous specific immunotherapy. Created using https://BioRender.com. In this study, blood was drawn from the patient according to this protocol.

**Figure 5 ijms-26-07511-f005:**
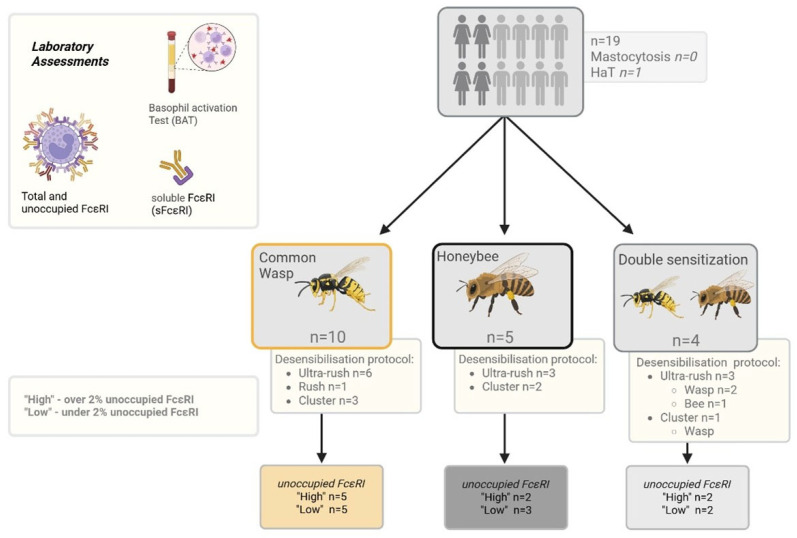
Patient grouping flowchart. HaT—hereditary alpha tryptasemia. Created using https://BioRender.com.

**Table 2 ijms-26-07511-t002:** Total and specific IgE comparisons for the “High” and “Low” unoccupied FcεRI groups.

	High Unoccupied FcεRI (*n* = 9)	Low Unoccupied FcεRI (*n* = 10)	*p*-Value
**Total IgE (U/mL)**	37 (49)	98.5 (126)	0.01 **
**Wasp sIgE (U/mL) ^a^**	2.0 (1) *(n = 6)*	2.0 (2) *(n = 7)*	0.39
**Bee sIgE (U/mL) ^a^**	10.5 (-) *(n = 3)*	6.7 (-) *(n = 3)*	0.83

^a^ The analysis of specific IgE (wasp/bee) was only performed for the group undergoing the respective venom immunotherapy (VIT). Statistical analysis was performed with a Mann–Whitney U test as the IgE values were abnormally distributed. Values are displayed as median (interquartile range). Significant difference is indicated as ** (*p*-value < 0.01).

**Table 3 ijms-26-07511-t003:** Delta of values before and after VIT for individuals with “High” (over 2%) or “Low” (under 2%) unoccupied FcεRI on basophils.

∆ Before–After VIT	High Unoccupied FcεRI (*n* = 9)	*p*-Value (Before/After)	Low Unoccupied FcεRI (*n* = 10)	*p*-Value (Before/After)	*p*-Value (Between the Groups)
**∆ Tryptase (µg/L)**	0.3 (1.35)	0.63	0.42 ± 0.52	0.03 *	0.55
**∆ Total FcεRI**	8212.97 ± 21,146.54	0.28	6373.87 ± 23,995.24	0.42	0.86
**∆ Unoccupied FcεRI**	299.27 (909.7)	0.44	−1224.43 ± 1083.77	<0.01 **	0.05 *
**∆ Ratio unoccupied/total FcεRI**	−0.003 (0.03)	0.14	−0.005 ± 0.004	<0.01 **	1.00
**∆ sFcεRI (ng/mL)**	0.138 ± 0.12	<0.01 **	0.093 ± 0.96	0.01 *	0.19

As the delta is calculated as before VIT minus after VIT, negative values indicate an increase. Statistical analysis was performed with a paired *t*-test or Wilcoxon test (if abnormally distributed) within the group or with either a *t*-test or Mann–Whitney test for group comparisons. Normally distributed values are depicted as median ± standard deviation; abnormally distributed values are displayed as median (interquartile range). VIT—venom immunotherapy; sFcεRI—soluble FcεRI. Significant is indicated as * (*p*-value <0.5) or ** (*p*-value < 0.01).

**Table 4 ijms-26-07511-t004:** Changes in BAT sensitivity correlated with FcεRI changes before/after VIT induction.

Correlation Coefficients	Wasp (*n* = 10)		Bee (*n* = 5)		All (*n* = 15)	
	∆ EC_50_ (µg/mL)	*p*-Value	∆ EC_50_ (µg/mL)	*p*-Value	∆ EC_50_ (µg/mL)	*p*-Value
**∆ Total FcεRI**	−0.733	0.02 *	−0.800	0.10	−0.768	<0.01 **
**∆ Unoccupied FcεRI**	−0.742	0.01 *	0.600	0.29	−0.104	0.71
**∆ Ratio of unoccupied/total FcεRI**	−0.292	0.41	0.800	0.10	0.268	0.33

Statistical analysis was performed with Spearman’s rho. Values shown are correlation coefficients and *p*-values. EC_50_—half-maximum basophil activation. Significant difference before and after VIT is indicated as * (*p*-value <0.5) or ** (*p*-value < 0.01).

**Table 5 ijms-26-07511-t005:** Cohort characteristics.

	HVA Patients (*n* = 19)
Age (y)	46.32 ± 15.1
Sex (f, %)	7 (36.8)
BMI (kg/m^2^)	26.53 ± 3.36
Mastocytosis (pos., %)	0 (0)
HaT (pos., %)	1 (5.3)
Other concomitant allergies (pos., %)	5 (35.71)
Total IgE (U/mL)	46.0 (76)
Hymenoptera sensibilization (*n*, %)	
Wasp	10 (52.6)
Bee	5 (26.3)
Both	4 (21.1)
Anaphylaxis (grade, %)	
I	2 (10.5)
II	14 (73.7)
III	3 (15.8)
IV	0 (0)
Desensitized venom (*n*, %)	
Wasp	13 (68.4)
Bee	6 (31.6)
Desensitization Protocol (*n*, %)	
Ultra-rush	12 (63.2)
Rush	1 (5.3)
Cluster	6 (31.6)

Results are shown as mean ± SD or median (IQR) (if abnormally distributed). y—Years; SD—standard deviation; f—female; BMI—body mass index; HaT—hereditary alpha tryptasemia; IQR—interquartile range.

## Data Availability

The datasets used and/or analyzed during this study are available from the corresponding author upon reasonable request.

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
