# Peer review of "Rapid Change in FcεRI Occupancy on Basophils After Venom Immunotherapy Induction"

_ijms, 2025, doi:10.3390/ijms26157511_

Round 1
Reviewer 1 Report
Comments and Suggestions for Authors
This manuscript explores early immune changes during venom immunotherapy in patients with hymenoptera allergy. The topic is relevant and the study is clearly described. The results are interesting, but some points need clarification and further discussion before publication.
- The sample size of 19 patients is relatively small, and further subgrouping (e.g., bee vs. wasp venom, high vs. low unoccupied FcεRI, different VIT protocols) reduces the statistical power even more. While the findings are interesting, the limited sample size may affect the reliability and generalizability of the results. It would be helpful if the authors could discuss this limitation in more detail and clarify whether the current sample is sufficient to support the subgroup analyses presented.
- The authors suggest on lines 243–247 that the observed changes in FcεRI expression may result from receptor internalization and recycling rather than de novo synthesis. While this is a reasonable hypothesis, the manuscript does not present direct experimental evidence to support this mechanism. I recommend adding references to relevant studies and clearly stating that this remains speculative without further validation.
- In Figure 1, statistical significance markers (e.g., * and **) are used, but their meaning is not clearly explained in the figure legend. Please include a note in the legend to define these symbols for clarity.
- In the Methods section (lines 346–349), the authors state that BAT results with ≥21% activation in the negative control were excluded due to high preactivation, and those with <40% activation in the positive control were considered non-responders. However, the rationale and references for choosing these cutoff values are not provided. Please include appropriate literature to support these thresholds or explain how they were determined.
Author Response
This manuscript explores early immune changes during venom immunotherapy in patients with hymenoptera allergy. The topic is relevant, and the study is clearly described. The results are interesting, but some points need clarification and further discussion before publication.
- Response: We thank the reviewer for the overall positive response.
- The sample size of 19 patients is relatively small, and further subgrouping (e.g., bee vs. wasp venom, high vs. low unoccupied FcεRI, different VIT protocols) reduces the statistical power even more. While the findings are interesting, the limited sample size may affect the reliability and generalizability of the results. It would be helpful if the authors could discuss this limitation in more detail and clarify whether the current sample is sufficient to support the subgroup analyses presented.
- Response: Thank you for highlighting this important limitation. We are aware that we only have a small study cohort. Of course, this also influences the generalizability of the results, but it is important for us to show the data in order to support future - perhaps more complex studies - with a larger cohort. To emphasis this limitation of the study we have extended the wording in the discussion section (Line 285-287).
- The authors suggest on lines 243–247 that the observed changes in FcεRI expression may result from receptor internalization and recycling rather than de novo synthesis. While this is a reasonable hypothesis, the manuscript does not present direct experimental evidence to support this mechanism. I recommend adding references to relevant studies and clearly stating that this remains speculative without further validation.
- Response: This is right, we do not have experimental evidence to support this hypothesis. We have added this information in the discussion and inserted respective references. (Line 253-256)
- In Figure 1, statistical significance markers (e.g., * and **) are used, but their meaning is not clearly explained in the figure legend. Please include a note in the legend to define these symbols for clarity.
- Response: Thank you for indicating this missing information. The explanation has been added. (Figure 1 and Figure 2)
- In the Methods section (lines 346–349), the authors state that BAT results with ≥21% activation in the negative control were excluded due to high preactivation, and those with <40% activation in the positive control were considered non-responders. However, the rationale and references for choosing these cutoff values are not provided. Please include appropriate literature to support these thresholds or explain how they were determined.
- Response: Thank you for pointing out this missing information. We have clarified this in the revised version of the manuscript. (Line 365-376):
Since no given cut-off for the negative control in the manufacturer manual (39) and no generally accepted cut-offs are established for the BAT, it had been suggested that the cut-off for the negative control and its coefficient of variation should be adopted by each laboratory (40). In the study lab the mean+/-SD of the negative control max. activation was 13.5% +/-8.5% and the cut off set at 21%.
The proposed cut-off for the positive control in the manufacturer manual (39) was given at >20%. However, since our laboratory set negative control was higher and the study aimed at following changes in the BAT over time (not just identifying sensitization), the cut off was set higher at 40% to better monitor potentially meaningful changes.
Over all two patients were excluded due to negative control exclusion and two due to positive control exclusion.
Reviewer 2 Report
Comments and Suggestions for Authors
The author investigated if the induction of VIT leads to alterations in the surface expression levels of FcεRI on basophils and claim it as a potential tools for monitoring treatment response in hymenoptera venom allergy management. The significance of this study is missing in the Introduction. The author presents brief description of the methods in the introduction part and defines few well established terms/ methods. This can be consolidated and brief information on the significance of the study can be provided.
There seems to be inconsistent results with BAT following VIT. Further studies with large sample should be considered to validate these results.
Has the author looked into Mast cells as FcεRI play an imp role in its activation/degranulation.
Method:
4.2 A paragraph has been italicized. The protocol for rush and cluster VIT needs further elaboration. Author presents timeline for both protocol but fails to describe in detail in the method section.
Figure.
Figure 1. Can the author update y-axes labelling with respective units.
Comments on the Quality of English Language
The manuscript needs to be proofread by a native English speaker as it contains many typing errors. Please throughly revise the manuscript.
Author Response
The author investigated if the induction of VIT leads to alterations in the surface expression levels of FcεRI on basophils and claim it as a potential tools for monitoring treatment response in hymenoptera venom allergy management. The significance of this study is missing in the Introduction. The author presents brief description of the methods in the introduction part and defines few well established terms/ methods. This can be consolidated and brief information on the significance of the study can be provided.
- Response: We thank the reviewer for the helpful comments. We have shortened the start of the introduction and added a few lines regarding the significance of the study at the end of the introduction (line 83-87)
There seems to be inconsistent results with BAT following VIT. Further studies with large sample should be considered to validate these results.
- Response: Thank you for pointing this out. We also think that a larger sample group is necessary to validate our data, as we point out in the discussion (Line 285-287)
Has the author looked into Mast cells as FcεRI play an imp role in its activation/degranulation.
- Response: Thank you for this valuable input, we also think this would be very interesting and important for future studies. Until now we did not look into the mast cells FcεRI, as mast cells are tissue resident (e.g. in the skin) cells and hard to harvest in individuals undergoing VIT. We have added this point into the aims for further studies (Line 291)
Method:
4.2 A paragraph has been italicized. The protocol for rush and cluster VIT needs further elaboration. Author presents timeline for both protocol but fails to describe in detail in the method section.
- Response: Thank you for spotting this formatting mistake and indicating this missing information. Further explanation on VIT protocols has been added. (Line 318-329)
Figure:
Figure 1. Can the author update y-axes labelling with respective unit
- Response: An updated Figure is now in the manuscript.
The manuscript needs to be proofread by a native English speaker as it contains many typing errors. Please thoroughly revise the manuscript.
Response: The manuscript was proofread and updated by a native English speaker.